# ThEBES: Thorough Energy-Based Evolution Strategy

## Abstract

Recently, Evolution Strategies (ESs) have achieved state-of-the-art results: ESs are a family of evolutionary algorithms that iteratively update the parameters of a search distribution to sample solutions to be evaluated. By optimizing a population, ESs promise to evolve solutions that are robust. Nevertheless, current methods have yet to deliver on this promise. We include an explicit drive towards robustness by applying noise to the search distribution mean after evaluating the solutions, adding a stochastic drift to the ES search trajectory. We mathematically ground our algorithm on Energy-Based Models (EBMs) and interpret it as performing Langevin dynamics on the search space, thus converging to a probability distribution and not a point estimate for the search distribution parameters. So we introduce ThEBES, the Thorough Energy-Based Evolution Strategy. We compare ThEBES against state-of-the-art ESs on continuous policy search tasks. Our results show that ThEBES is competitive in terms of effectiveness. We also find that, by virtue of its stochastic dynamics, ThEBES evolves policies that are more robust to observational noise. We thus believe our work to be a promising avenue for future research and to strengthen the theoretical backings of ESs, since it provides a solid mathematical ground to ESs in the context of energy-based models.

## 1 Introduction

Evolution Strategies (ESs) (Rechenberg, 1973) are a family of evolutionary algorithms (De Jong, 2006) that iteratively update the parameters of a search distribution to sample a population of solutions to be evaluated. They have recently achieved state-of-the-art performance on continuous policy search tasks (Chrabaszcz et al., 2018) for reinforcement learning agents and are competitive to established value-based methods (Salimans et al., 2017; Such et al., 2018). But, performance by itself may not be strictly sufficient, as we seek agents that are *robust*: a small perturbation to the agent's observations must correspond to a small perturbation to the agent's behavior (Kirk et al., 2023). Robustness plays a key role in agents that can adapt to their environment and interact safely with other agents, including humans, but most current reinforcement learning agents are embarrassingly frail (Cobbe et al., 2019). By evaluating a "cloud" of solutions, ESs promise to evolve solutions that are robust to perturbations in the parameter space, so that they are likely to be robust to perturbations in the observation space. Nevertheless, empirical evidence suggests that current methods have yet to deliver on this promise (Chrabaszcz et al., 2018; Lehman et al., 2018; Such et al., 2018): evolved solutions are usually as frail as their value-based counterparts and fail in terms of robustness.

Here we include an explicit drive towards robustness by applying noise to the search distribution mean after evaluating the solutions, a process we call "smearing"; as a result, the evolution of the solutions follows a more stochastic trajectory in the search space, covering a wider cloud of points. We interpret this approach as performing Langevin dynamics—a Bayesian method that adds a stochastic drift term to gradient ascent—on the search space. Langevin dynamics converge to a probability distribution and not a point estimate for the search distribution parameters, an approach that falls in the domain of Energy-Based Models (EBMs) (LeCun et al., 2006). We thus introduce the Thorough Energy-Based Evolution Strategy (ThEBES).

We compare ThEBES against other state-of-the-art ESs on four continuous policy search tasks and find it to be competitive. Chiefly, we also test the robustness of the evolved solutions empirically, by perturbing the state observations with noise. We show that ThEBES achieves higher robustness

to observational noise when compared to the other methods. Our results suggest that ThEBES evolves solutions that are comparable in regular conditions, but better in noisy conditions, pointing to interesting directions for future work. In particular, we showcase its potential for transfer learning without fine-tuning: we collect an ensemble of solutions by sampling the population at different stages of evolution and find that ensembles evolved through ThEBES are diverse enough to adapt to new environments.

Finally, ThEBES strengthens the theoretical backings of ESs, since it provides a solid mathematical ground to ESs in the context of EBMs, by showing that the fitness function can be transformed into an energy function to optimize. Such a contribution is relevant in its own right, since researchers have historically struggled to provide mathematical foundations to evolutionary algorithms (De Jong, 2006).

## 2 BACKGROUND AND RELATED WORK

Before delving into the derivation of ThEBES, we present the background on ESs and energy-based modeling. Let $\mathcal{X} \subseteq \mathbb{R}^n$ be a numerical $n$-dimensional solution space, consisting of solutions to an optimization problem (e.g., parameters for neural network controllers for reinforcement learning agents). Let $f : \mathcal{X} \to \mathbb{R}$ be a fitness function (without loss of generality, to be maximized), telling the quality of any of the solutions, that we do not make any assumptions about; so, $f$ is a black-box function. Our black-box optimization problem consists in finding:

$$\arg\max_{\boldsymbol{x} \in \mathcal{X}} f(\boldsymbol{x}) \tag{1}$$

### 2.0.1 EVOLUTION STRATEGIES

One relevant instance of black-box optimization algorithms is ESs (Hansen et al., 2015), a family of evolutionary algorithms (De Jong, 2006). For example, ESs have achieved state-of-the-art results for continuous control (Salimans et al., 2017) and game-playing (Chrabaszcz et al., 2018). Moreover, they are easy to parallelize and generally robust to hyperparameter settings (Heidrich-Meisner & Igel, 2008). First introduced by Rechenberg (1973), ESs have since flourished, with new advancements that include invariance principles (Ollivier et al., 2017) and meta-learning (Lange et al., 2023).

While ESs encompass many particular algorithms, Wierstra et al. (2014) introduced a popular class that evolves a fixed-size population—consisting of individuals $\boldsymbol{x}_i$—of $\lambda$ solutions as samples from a search distribution, itself parametrized by $\boldsymbol{\theta} \in \Theta \subseteq \mathbb{R}^p$. The search distribution is set to be an isotropic Gaussian $N(\boldsymbol{\theta}, \sigma^2 \boldsymbol{I})$, where $\sigma$ is the step-size. The parameters of the search distribution consist only of the mean $\boldsymbol{\theta}$. Then, the problem of Equation (1) is recast as an optimization in the $p$-dimensional search space $\Theta$:

$$\arg\max_{\boldsymbol{\theta} \in \Theta} \mathbb{E}_{\boldsymbol{x} \sim N(\boldsymbol{\theta}, \sigma^2 \boldsymbol{I})} \left[ f(\boldsymbol{x}) \right] \tag{2}$$

where we take the expected value of $f$ over the population of sampled solutions because the $\boldsymbol{x}_i$ are realizations of a search distribution $N(\boldsymbol{\theta}, \sigma^2 \boldsymbol{I})$ (in other words, $\boldsymbol{x}_i = \boldsymbol{\theta} + \sigma \boldsymbol{\epsilon}_i$, with $\boldsymbol{\epsilon}_i \sim N(\boldsymbol{0}, \sigma^2 \boldsymbol{I})$). We present the pseudo-code for such an ES in Algorithm 1; our goal is not to propose a comprehensive formulation of ESs, but to lay the framework for our method (see Section 3).

At every iteration $t$, ES takes a step (damped by learning rate $\alpha$) in the direction of the gradient of Equation (2) $\nabla_{\boldsymbol{\theta}} \mathbb{E}_{\boldsymbol{x} \sim N(\boldsymbol{\theta}, \sigma^2 \boldsymbol{I})} \left[ f(\boldsymbol{x}) \right]$ (line 9), which has been proven to approximate the expression on the right-hand side of line 8: $\frac{1}{\sigma \lambda} \sum_{i=1}^{\lambda} s_i \boldsymbol{\epsilon}_i$ and we include the proof in Appendix A.1.

In other words, ES iteratively updates the search distribution mean as the weighted recombination of the current population, where the weights are the fitness values and thus favor fitter solutions. This process continues until $n_{\text{evals}}$ fitness evaluations have been computed and outputs the mean at the last iteration $\boldsymbol{\theta}^*$.

### 2.1 ENERGY-BASED MODELS

ESs take a generative modeling approach: they aim to estimate the density of high-fitness solutions through a search distribution, whose parameters are to be adapted. By iteratively updating $\boldsymbol{\theta}_t$ with

**input :** $n_{\text{evals}}$ number of fitness evaluations, $\lambda$ population size, $\boldsymbol{\theta}_0$ initial search distribution mean, step-size $\sigma$, fitness function $f$, learning rate $\alpha$.
**output:** $\boldsymbol{\theta}^*$ search distribution mean at the last iteration.

1   $t \leftarrow 0$
2   $i \leftarrow 0$
3   **while** $i < n_{evals}$ **do**
4     $\boldsymbol{\epsilon}_1, \ldots, \boldsymbol{\epsilon}_\lambda \sim N(\mathbf{0}, \sigma^2 \boldsymbol{I})$
5     **foreach** $i \in \{1, \ldots, \lambda\}$ **do**
6       $s_i \leftarrow f(\boldsymbol{\theta}_t + \sigma \boldsymbol{\epsilon}_i)$
7     **end**
8     $\boldsymbol{g}_t \leftarrow \frac{1}{\sigma \lambda} \sum_{i=1}^{\lambda} s_i \boldsymbol{\epsilon}_i$
9     $\boldsymbol{\theta}_{t+1} \leftarrow \boldsymbol{\theta}_t - \alpha \boldsymbol{g}_t$
10    $t \leftarrow t + 1$
11    $i \leftarrow i + \lambda$
12 **end**
13 $\boldsymbol{\theta}^* \leftarrow \boldsymbol{\theta}_t$

**Algorithm 1:** The pseudo-code for ES.

steps in the direction of the gradient of $\mathbb{E}_{\boldsymbol{x} \sim N(\boldsymbol{\theta}_t, \sigma^2 \boldsymbol{I})}[f(\boldsymbol{x})]$, ESs converge to a point estimate for $\boldsymbol{\theta}_t$; in other words, they perform a maximum likelihood estimation of the search distribution parameters. In this way, ESs miss the opportunity to model a more complex distribution of potential solutions. EBMs (LeCun et al., 2006), a class of generative models, are general enough to comprise maximum likelihood models but are less restrictive. Whereas maximum likelihood converges to a point estimate for $\boldsymbol{\theta}$, EBMs allow us to take into consideration the uncertainty surrounding $\boldsymbol{\theta}$.

EBMs have recently come under the spotlight for generative modeling (Song & Ermon, 2019), propelled by a tractable sampling procedure (Welling & Teh, 2011). Notably, Che et al. (2020) found that formulating generative adversarial networks—a popular class of neural generative models—as EBMs led to more efficient training; similarly, Grathwohl et al. (2019) proved that the same holds true for standard classifiers. From the policy search perspective, Haarnoja et al. (2017) incorporated EBMs into established reinforcement learning methods to learn repertoires of behaviors; Du & Mordatch (2019) highlighted better generalization for EBMs, while Henaff et al. learned policies from purely observational data with no environment interactions.

In detail, we want to estimate a posterior probability $p_{\boldsymbol{x} \sim N(\boldsymbol{\theta}, \sigma^2 \boldsymbol{I})} : \Theta \to [0, 1]$. $p_{\boldsymbol{x} \sim N(\boldsymbol{\theta}, \sigma^2 \boldsymbol{I})}$ is the probability that, after observing $\boldsymbol{x}$ according to $N(\boldsymbol{\theta}, \sigma^2 \boldsymbol{I})$, $\boldsymbol{\theta}$ is $\boldsymbol{\theta}$. In most scenarios, we are not in the condition to make any assumptions on $p_{\boldsymbol{x} \sim N(\boldsymbol{\theta}, \sigma^2 \boldsymbol{I})}$. But, we can express—through Boltzmann's trick—any probability distribution as:

$$p_{\boldsymbol{x} \sim N(\boldsymbol{\theta}, \sigma^2 \boldsymbol{I})}(\boldsymbol{\theta}) = \frac{\exp\left(-E_{\boldsymbol{x} \sim N(\boldsymbol{\theta}, \sigma^2 \boldsymbol{I})}(\boldsymbol{\theta})\right)}{z(\boldsymbol{x})} \tag{3}$$

where $E : \Theta \to \mathbb{R}$ is an *energy function*, the exponentiation ensures the probabilities are non-negative, and dividing by $z(\boldsymbol{x}) = \int_{\boldsymbol{x} \in \mathcal{X}} \exp\left(-E_{\boldsymbol{x} \sim N(\boldsymbol{\theta}, \sigma^2 \boldsymbol{I})}(\boldsymbol{\theta})\right) d\boldsymbol{x}$ (the partition function) ensures the probabilities lie in $[0, 1]$. There are no assumptions on the energy function: it can be any function, provided that it maps to a scalar.

EBMs converge to an estimate for $p_{\boldsymbol{x} \sim N(\boldsymbol{\theta}, \sigma^2 \boldsymbol{I})}$ by optimizing (via, e.g., gradient ascent) the right-hand side of Equation (3). But, for most energy functions, the partition function is intractable as it requires integrating over the set of all possible $\boldsymbol{x}$, which historically limited the development of EBMs. Following (Welling & Teh, 2011), we can approximate a gradient ascent on the energy by means of Langevin dynamics (Namiki, 2008) (an iterative optimization scheme) since, as we shall see, they transform our problem into a tractable one and converge to a posterior distribution for the parameters $\boldsymbol{\theta}$. Indeed, Langevin dynamics originally developed as a discretization of a stochastic differential equation whose equilibrium distribution is the posterior distribution; in physics, Langevin dynamics describe the time evolution of particles that are subject to both deterministic and stochastic forces.

Langevin dynamics modify the usual gradient ascent update by injecting noise such that the trajectory of the parameters converges to the full posterior distribution rather than just a point estimate.

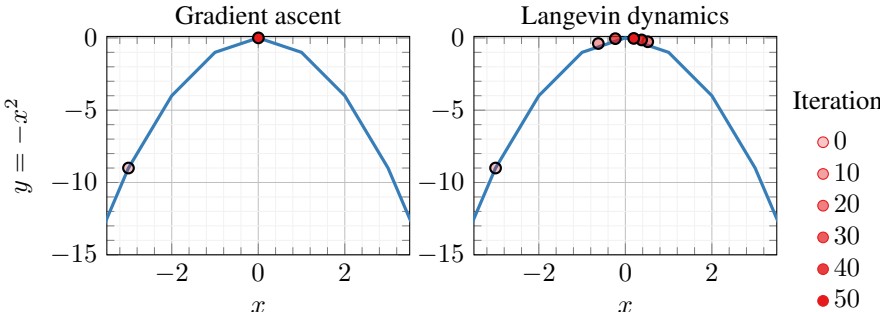

Figure 1: Path for gradient ascent and Langevin dynamics on the toy function $y = -x^2$ (so $\log p_{\boldsymbol{x} \sim N(\boldsymbol{\theta}, \sigma^2 \boldsymbol{I})}(\boldsymbol{\theta}) = -x^2$). Because of the stochastic drift in the update, the latter does not ascend straight to the maximum but bounces back and forth until settling on it. For this test, $\alpha = 0.1$, $\theta_0 = 3$, and $\sigma = 1$.

They unfold as follows:

$$\boldsymbol{\theta}_{t+1} = \boldsymbol{\theta}_t + \alpha \nabla_{\boldsymbol{\theta}} \log p_{\boldsymbol{x} \sim N(\boldsymbol{\theta}, \sigma^2 \boldsymbol{I})}(\boldsymbol{\theta}) + \sqrt{\alpha} \sigma \boldsymbol{\eta}_t \tag{4}$$

$$\boldsymbol{\eta}_t \sim N(\boldsymbol{0}, \boldsymbol{I}) \tag{5}$$

where $\boldsymbol{\eta}_t$ is a stochastic drift. We illustrate the difference between gradient ascent and Langevin dynamics in Figure 1 for a 1-dimensional function. Because of the stochastic drift, Langevin dynamics do not ascend straight to the maximum but bounce back and forth until settling on it.

Whereas Welling & Teh (2011) does not refer to energy-based modeling when introducing Langevin dynamics, EBMs allow us to link Langevin dynamics to ESs, as we shall see in the next section.

## 3 THEBES: THOROUGH ENERGY-BASED EVOLUTION STRATEGY

We introduce ThEBES, starting from the algorithmic formulation, then moving on to the theoretical explanation. We present the pseudo-code in Algorithm 2: it consists of the generic ES of Algorithm 1 with the addition of stochastic drift to the gradient update at line 10, a process that we call "smearing" (because it perturbs the search distribution mean). In other words, we perform a Langevin (and not gradient) ascent in the parameter space: by virtue of the stochastic drift, we evolve solutions that are more robust to observational noise.

**input** : $n_{\text{evals}}$ number of fitness evaluations, $\lambda$ population size, $\boldsymbol{\theta}_0$ initial search distribution mean, step-size $\sigma$, fitness function $f$, learning rate $\alpha$.
**output:** $\theta^*$ search distribution mean at the last iteration.

1  $t \leftarrow 0$
2  $i \leftarrow 0$
3  **while** $i < n_{evals}$ **do**
4  $\quad \boldsymbol{\epsilon}_1, \dots, \boldsymbol{\epsilon}_\lambda \sim N(\boldsymbol{0}, \sigma^2 \boldsymbol{I})$
5  $\quad$ **foreach** $i \in \{1, \dots, \lambda\}$ **do**
6  $\quad\quad s_i \leftarrow f(\boldsymbol{\theta}_t + \sigma \boldsymbol{\epsilon}_i)$
7  $\quad$ **end**
8  $\quad \boldsymbol{g}_t \leftarrow \frac{1}{\sigma \lambda} \sum_{i=1}^{\lambda} s_i \boldsymbol{\epsilon}_i$
9  $\quad \boldsymbol{\eta}_t \sim N(\boldsymbol{0}, \boldsymbol{I})$
10  $\quad \boldsymbol{\theta}_{t+1} \leftarrow \boldsymbol{\theta}_t + \alpha \boldsymbol{g}_t + \sqrt{\alpha} \sigma \boldsymbol{\eta}_t$
11  $\quad t \leftarrow t + 1$
12  $\quad i \leftarrow i + \lambda$
13  **end**
14  $\boldsymbol{\theta}^* \leftarrow \boldsymbol{\theta}_t$

**Algorithm 2:** The pseudo-code for ThEBES.

We here provide a mathematically-grounded explanation for our formulation. We believe that ThEBES does not converge to a point estimate for the parameters $\boldsymbol{\theta}$, but to the posterior density $p_{\boldsymbol{x} \sim N(\boldsymbol{\theta}, \sigma^2 \boldsymbol{I})}$ (mentioned in Section 2.1), thus taking a Bayesian perspective.

We begin from the ES of Algorithm 1. As discussed in Section 2.1, Langevin dynamics provide the tool to converge to a posterior distribution for $\boldsymbol{\theta}$. Nevertheless, we shall not rely on Equation (4) as it is, since we still need a way to compute $p_{\boldsymbol{x} \sim N(\boldsymbol{\theta}, \sigma^2 \boldsymbol{I})}(\boldsymbol{\theta})$ before approximating $\nabla_{\boldsymbol{\theta}} \log p_{\boldsymbol{x} \sim N(\boldsymbol{\theta}, \sigma^2 \boldsymbol{I})}(\boldsymbol{\theta})$ via sampling a population. So, we now derive a more tractable form for it. Through Boltzmann's trick (Equation (3)), we can express $p_{\boldsymbol{x} \sim N(\boldsymbol{\theta}, \sigma^2 \boldsymbol{I})}(\boldsymbol{\theta})$ as the function of an energy $-E_{\boldsymbol{x} \sim N(\boldsymbol{\theta}, \sigma^2 \boldsymbol{I})}(\boldsymbol{\theta})$. We then find that:

$$\nabla_{\boldsymbol{\theta}} \log p_{\boldsymbol{x} \sim N(\boldsymbol{\theta}, \sigma^2 \boldsymbol{I})}(\boldsymbol{\theta}) = \nabla_{\boldsymbol{\theta}} \left( \log \exp \left( -E_{\boldsymbol{x} \sim N(\boldsymbol{\theta}, \sigma^2 \boldsymbol{I})}(\boldsymbol{\theta}) \right) - \log z(\boldsymbol{x}) \right)$$
$$= -\nabla_{\boldsymbol{\theta}} E_{\boldsymbol{x} \sim N(\boldsymbol{\theta}, \sigma^2 \boldsymbol{I})}(\boldsymbol{\theta}) \qquad (6)$$

because the partition function does not depend on $\boldsymbol{\theta}$. In synthesis, we are formulating our problem as that of ascending the landscape of an energy function (and the method to do so is exactly Langevin dynamics). $-\mathbb{E}_{\boldsymbol{x} \sim N(\boldsymbol{\theta}, \sigma^2 \boldsymbol{I})}[f(\boldsymbol{x})]$ from Equation (2), whose gradient we already derived as $\frac{1}{\sigma \lambda} \sum_{i=1}^{\lambda} f(\boldsymbol{x}_i) \boldsymbol{\epsilon}_i$, is a function of $\boldsymbol{\theta}$, depends on $\boldsymbol{x}$, and it maps to a scalar: it is thus a suitable candidate for the energy. In other words, the energy is a smoothing of the fitness function that makes the fitness landscape smoother and hence better behaved.

We can eventually plug this result back into Equation (4) to obtain the following update rule:

$$\boldsymbol{\theta}_{t+1} = \boldsymbol{\theta}_t + \frac{\alpha}{\sigma \lambda} \sum_{i=1}^{\lambda} f(\boldsymbol{x}_i) \boldsymbol{\epsilon}_i + \sqrt{\alpha} \sigma \boldsymbol{\eta}_t \quad (7)$$

$$\boldsymbol{\eta}_t \sim N(\boldsymbol{0}, \boldsymbol{I}) \qquad (8)$$

As a result of the stochastic drift $\boldsymbol{\eta}_t$, $\boldsymbol{\theta}_{t+1} \sim N(\boldsymbol{\theta}_t + \frac{\alpha}{\sigma \lambda} \sum_{i=1}^{\lambda} f(\boldsymbol{x}_i) \boldsymbol{\epsilon}_i, \alpha \sigma^2 \boldsymbol{I})$ and it is no longer a point estimate. We illustrate the difference between plain and smeared gradient in Figure 2.

We introduce the Thorough Energy-Based Evolution Strategy (ThEBES ): "Thorough" because it follows a rigorous mathematical derivation; "Energy-Based" because it converges to a posterior distribution for the search parameters $\boldsymbol{\theta}$; "Evolution Strategy" because it belongs to this family of algorithms. With respect to the ES of Algorithm 1, it amounts to a simple modification: the addition of stochastic drift (smearing) at line 10. In a world that teems with bulky and intricate algorithms, ThEBES is an elegant addition to consolidated knowledge and does not require further hyperparameters.

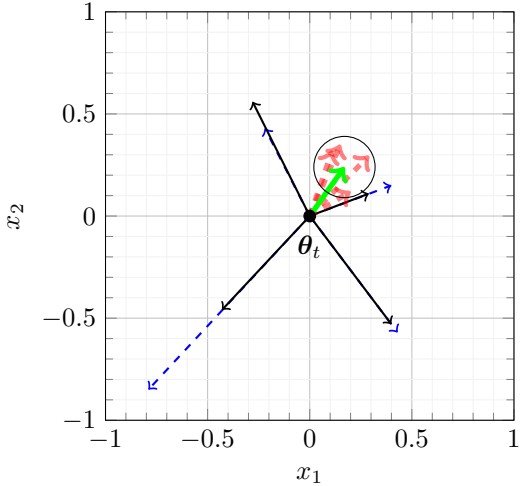

Figure 2: Geometric interpretation of plain and smeared gradient on a toy 2-dimensional space. Black arrows are the $\boldsymbol{\epsilon}_i$ perturbations, blue arrows are the $f(\boldsymbol{x}_i) \boldsymbol{\epsilon}_i$, the green arrow is the plain gradient $\sum_{i=1}^{\lambda} f(\boldsymbol{x}_i) \boldsymbol{\epsilon}_i$, while the red arrows are the smeared gradient $\sum_{i=1}^{\lambda} f(\boldsymbol{x}_i) \boldsymbol{\epsilon}_i + \sqrt{\alpha} \sigma \boldsymbol{\eta}_t$ under different sampled perturbations $\boldsymbol{\eta}_t$.

## 4 EXPERIMENTAL EVALUATION

Our goal is to investigate the following research questions:

RQ1 Is ThEBES effective when compared to other established algorithms?

RQ2 If yes, why does ThEBES work? In particular, does the stochasticity in the gradient update make the evolved solutions more robust?

We investigate these questions on reinforcement learning agents, because of the relevance that robustness plays in the adaptation to environmental changes. In particular, we consider four

continuous policy search tasks (see **??**), namely `CartPole-Hard`, `HalfCheetah`, `Ant`, and `LunarLander`. We compare ThEBES with state-of-the-art ESs (see Section 4.1). For all the tasks, the policy $x$ consists of the parameters (i.e., weights and biases) of a feed-forward, fully-connected neural network controller for the agents of the form: $a_k = h_x(o_t)$, where $a \in \mathbb{R}^{n_{\text{out}}}$ is the output (the action for the agent) and $o \in \mathbb{R}^{n_{\text{in}}}$ is the observation input to the controller at time step $t$. Unless otherwise specified, the neural network has 2 hidden layers of 64 neurons each and $\tanh$ as the activation function. As a result of its action, the agent receives a reward $r_k \in \mathbb{R}$ at time step $k$ to accrue the final fitness: $f = \sum_{t=1}^{t_{\text{final}}} r_k$, where $t_{\text{final}}$ is the total number of time steps in the simulation.

We include the hyperparameter settings for all algorithms in Appendix A.4. Moreover, we experiment and discuss the impact of selected hyperparameters on ThEBES in Appendix A.5.

For every experiment, we performed 5 independent runs varying the random seed. For a given seed and policy, every fitness evaluation is deterministic. We carried out the statistical tests using the Mann Whitney U-rank test for independent samples. The code is publicly available at *URL omitted for double-blind review*. We ran all experiments on a Google Colab Pro notebook equipped with an A100 Nvidia GPU. Each run took approximately $25\,\text{s}$ for `CartPole-Hard` and $500\,\text{s}$ for `HalfCheetah`, `Ant`, and `LunarLander`.

## 4.1 BASELINES

As baselines, we consider:

**OpenAI-ES (OpenES) (Salimans et al., 2017):** according to (Pagliuca et al., 2020), it is one of the most effective modern ESs and proved competitive to state-of-the-art reinforcement learning algorithms (Salimans et al., 2017). OpenES is algorithmically similar to ThEBES, with the exception of the stochastic drift, making it an ideal baseline. Moreover, we apply the same implementation details of Appendix A.2. With respect to the original implementation, this one does not employ weight decay and virtual batch normalization.

**Random Search (RS):** it is a pure random search algorithm; as a result, it is an ideal minimum baseline. It uniformly samples individuals from $[-5, 5]^n$ and returns the highest-fitness one.

**Covariance Matrix Adaptation Evolution Strategy (CMA-ES) (Hansen & Ostermeier, 2001):** an established numerical optimization algorithm. CMA-ES iteratively optimizes the solution in the form of a multivariate normal distribution. At each iteration, it samples the distribution obtaining a population of individuals and then updates the parameters of the distribution based on the best half of the population. CMA-ES employs non-trivial heuristics while updating the distribution—we refer the reader to Hansen & Ostermeier (2001).

**Separable Natural Evolution Strategy (sNES) (Wierstra et al., 2014):** an established instance of natural evolution strategies, which adopt probabilistic models to estimate the gradient of the fitness function. In addition, sNES adapts the step-size to strike a balance between exploration and exploitation.

**Adaptive Sample Efficient Black-box Optimization (ASEBO) (Choromanski et al., 2019):** ASEBO adapts to the geometry of the fitness function by dynamically learning the intrinsic dimensionality of the gradient. In experiments on continuous control tasks, it led to better sample-efficiency than state-of-the-art ESs.

We implemented ThEBES using EvoJAX (Tang et al., 2022), a recent GPU-accelerated evolutionary framework. For the baselines, we adopted their EvoJAX implementation when available, otherwise we reimplemented their EvoSAX (Lange, 2023) one into the framework. For `LunarLander`, that is not available in EvoJAX, we reimplemented all the algorithms using the reference official implementation released by the respective authors.

## 4.2 RESULTS

We present and analyze the experimental results of our research questions.

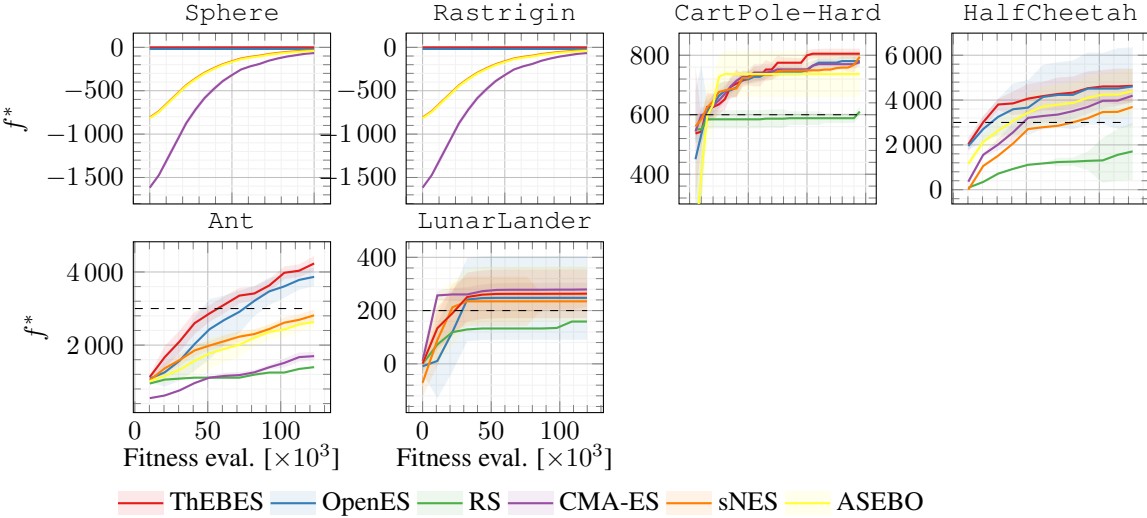

Figure 3: Median $\pm$ standard deviation (solid line and shaded area) for the fitness of the best policies $f^*$ drawn from 5 independent runs. The dashed lines stand for the threshold for the tasks to be said "solved" according to (Tang et al., 2022; Brockman et al., 2016). ThEBES is competitive with other established algorithms on these tasks.

### 4.2.1 RQ1: IS THEBES EFFECTIVE WHEN COMPARED TO OTHER ESTABLISHED ALGORITHMS?

To establish the effectiveness of ThEBES as a direct policy search algorithm, we compare it against the baselines of Section 4.1. As an index of effectiveness, we consider the fitness of the best policy $f^*$. We report the results in terms of median $\pm$ standard deviation in Figure 3 as a function of evolutionary time; for each task, we highlight the "solved" threshold (Tang et al., 2022; Brockman et al., 2016) with a dashed black line. As an additional test, we include two functions from the Nevergrad (Bennet et al., 2021) black-box optimization library: `Sphere` and `Rastrigin`.

ThEBES evolves policies that solve the tasks and are competitive with other approaches. In case of Nevergrad, ThEBES manages to recover the optimum. With the exception of the `Ant` task, all lines settle on a plateau, and continuing evolution would unlikely bring any improvements. The best policies evolved by ThEBES are significantly better ($p < 0.01$) than those found by RS, a random search algorithm, that does not even succeed in solving the tasks of `HalfCheetah`, `Ant`, and `LunarLander`. Moreover, the evolved best policies of ThEBES are comparable to those found by the other state-of-the-art algorithms: their median $f^*$ is actually the highest in the `CartPole-Hard`, `Ant`, and `LunarLander` tasks, but the $p$-values are not significant. The results for all the baselines are in line with what was reported in their respective papers.

Through that evidence, we can positively answer to RQ1: ThEBES evolves policies that are effective when compared to other established algorithms.

### 4.2.2 RQ2: ARE THE POLICIES EVOLVED BY THEBES MORE ROBUST?

We hypothesize the reason for the effectiveness of ThEBES to be that, thanks to its stochastic drift component, it evolves policies that are more robust to observational noise. We select the best policy from each of the 5 independent runs of Section 4.2.1 and re-assess it on 100 independent test evaluations (with predefined random seeds) characterized by varying environmental conditions: for `CartPole-Hard`, the initial cart position, cart velocity, and pole angle are sampled uniformly in $[-1, 1]$ (the domain of the observations), while for `HalfCheetah` and `Ant` the initial joint positions are perturbed with noise uniformly distributed in $[-0.1, 0.1]$ and the velocities are sampled uniformly from $[-0.1, 0.1]$. As the performance index, we compute the average of $f$ over the re-assessment evaluations $f^*_{\text{test}}$. We report the $f^*_{\text{test}}$ distribution in Figure 4, together with the $p$-values in brackets.

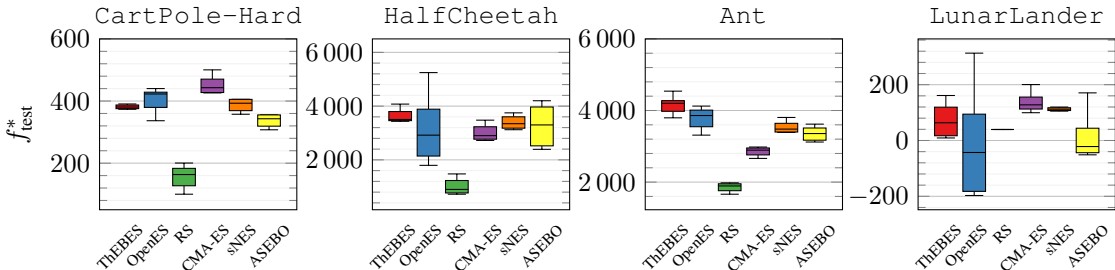

Figure 4: Distribution of the average fitness over 100 independent test evaluations with varying environmental conditions of the best policies drawn from 5 independent runs. ThEBES evolves policies that are generally more robust.

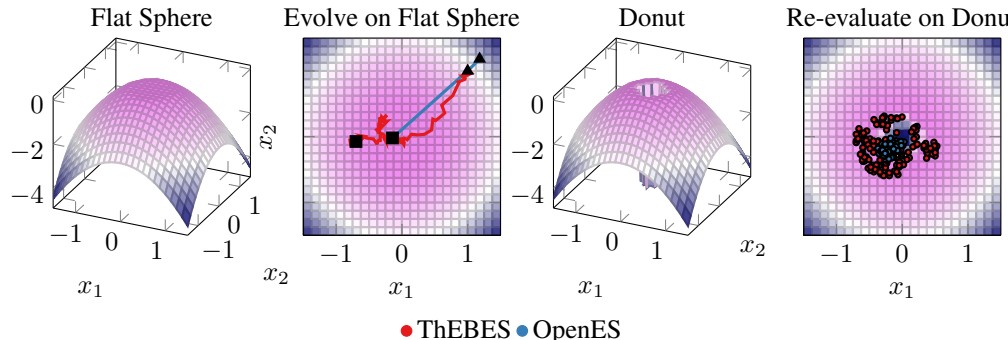

Figure 5: Left: evolutionary paths (solid lines) of $\boldsymbol{\theta}_t$ for one exemplar run on the 2-dimensional flat sphere function with a plateau centered at $(0, 0)$. The starting points are black diamonds, while the endings are black squares. Left: ThEBES traces a more stochastic path and hovers around the plateau. Right: best individuals taken at every 10 iteration over 5 runs, plotted over the donut. ThEBES individuals still cover a wide area, while most OpenES individuals fall into the donut hole.

ThEBES evolves policies that are generally more robust than those found by the baselines. Its evolved policies are even significantly better than those found by the baselines ($p < 0.05$), with the exception of CMA-ES on the `CartPole-Hard` and `LunarLander` and sNES on `LunarLander`.

To better visualize this intuition, we plot in Figure 5 the evolutionary path (trajectory of $\boldsymbol{\theta}_t$, the red solid line) on the 2-dimensional flat sphere function: $f_{\text{sphere}}(\boldsymbol{x}) = -\max(x_1^2 + x_2^2, 0.25)$. $f_{\text{sphere}}$ is the simplest function that can be visualized in two dimensions since it is convex and smooth, and has one fitness plateau centered at $(0, 0)$; in this way, the posterior distribution should not concentrate on one optimum point but on the whole plateau. In the second plot from the left of Figure 5, we compare the path for one exemplar run of ThEBES and one exemplar run of OpenES, both consisting of $n_{\text{evals}} = 40\,000$; paths for different runs were similar. We adopted the same hyperparameters for `CartPole-Hard`.

The evolutionary paths of ThEBES and OpenES radically differ. OpenES traces a straight path toward the plateau, whereas ThEBES traces a more stochastic path. Intuitively, ThEBES zigzags because of the smearing component in the gradient update. We also remark that ThEBES ends up "wandering" over the plateau, thus better covering the posterior.

To elucidate the benefits of following a stochastic path, we carry out a transfer learning experiment on a toy function. We repeat the aforementioned experiments on the flat sphere for 5 independent runs and save the best individual every 10 iterations. We then re-evaluate these individuals on the donut function (third plot from the left of Figure 5) and visualize each one of them as a point (fourth plot from the left of Figure 5). The donut function is the flat sphere function with a hole of fitness $-4$ (of radius 0.25) at the center. The donut is thus a deceptive landscape, as we would expect any

individual that reached the optimum to likely fall in the hole. As we can see, the individuals evolved with ThEBES form a cloud scattered over a wide area around the fitness plateau, whereas individuals evolved with OpenES concentrate on the hole. The fitness is significantly different between the two algorithms ($p < 0.01$) for this new function, suggesting that ThEBES implicitly provides an ensemble of solutions that is diverse enough to adapt to new environments.

Through that evidence, we can positively answer to RQ2: ThEBES generally evolves policies that are more robust than those found by the baselines. Moreover, the results on toy functions suggest that ThEBES may offer transfer learning without fine-tuning, as it collects a more diverse set of solutions over its trajectory.

## 5 DISCUSSION AND CONCLUSION

Evolution Strategies (ESs) have achieved state-of-the-art results: by optimizing a population, they promise to evolve solutions that are robust. Nevertheless, current methods have yet to deliver on this promise (Chrabaszcz et al., 2018). We introduce the Thorough Energy-Based Evolution Strategy (ThEBES), a novel evolutionary algorithm (De Jong, 2006) that includes an explicit drive for robustness by perturbing the search distribution mean with noise. ThEBES is mathematically grounded and we derive the interpretation from Energy-Based Models (EBMs) (LeCun et al., 2006), as it approximates Langevin dynamics on the search space. We show that ThEBES achieves better robustness—when compared to state-of-the-art ESs—to observational noise on four continuous policy search tasks. We also suggest that ensembles of solutions evolved through ThEBES have the potential to adapt well to unseen environments.

Finally, ThEBES establishes a solid mathematical grounding to the connection between ESs and EBMs, arguing that the fitness function can be transformed into an energy function to optimize. Such a contribution is relevant in its own right, since researchers have historically struggled to provide mathematical foundations to evolutionary algorithms (De Jong, 2006).

### 5.1 LIMITATIONS

For our claims to be universal, more experiments are required, considering different types of robustness. We considered test environments coming from the same distribution of training environments, while an overarching evaluation of robustness would also consider out-of-distribution environments. Moreover, we here focused on observational noise, while it would also be interesting to consider robustness to parameter noise (i.e., perturbations in the parameter space (Such et al., 2018)).

### 5.2 BROADER IMPACT

Our algorithm can be applied in ways that may have potential negative impacts on the broader society. While the experiments of this paper consider self-contained simulated agents, future—albeit distant—agents may evolve harmful adaptation abilities. Still, we believe this scenario belongs to the world of science fiction for the foreseeable future.

ACHKNOWLEDGMENTS

*Omitted for double-blind review*

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

# A  APPENDIX

## A.1  PROOF OF LINE 8 IN ALGORITHM 1

We now prove that $\nabla_{\boldsymbol{\theta}} \mathbb{E}_{\boldsymbol{x} \sim N(\boldsymbol{\theta}, \sigma^2 \boldsymbol{I})} [f(\boldsymbol{x})] \approx \frac{1}{\sigma \lambda} \sum_{i=1}^{\lambda} s_i \boldsymbol{\epsilon}_i$, for the sake of Algorithm 1. By virtue of the policy gradient theorem (Williams, 1992), the following equality holds:

$$\nabla_{\boldsymbol{\theta}_t} \mathbb{E}_{\boldsymbol{x} \sim N(\boldsymbol{\theta}_t, \sigma^2 \boldsymbol{I})} [f(\boldsymbol{x})] = \mathbb{E}_{\boldsymbol{x} \sim N(\boldsymbol{\theta}_t, \sigma^2 \boldsymbol{I})} [f(\boldsymbol{x}) \nabla_{\boldsymbol{\theta}_t} \log N(\boldsymbol{\theta}_t, \sigma^2 \boldsymbol{I})] \tag{9}$$

The parameters of the search distribution consist only of the mean $\boldsymbol{\theta}_t$ and we have:

$$\nabla_{\boldsymbol{\theta}_t} \log N(\boldsymbol{\theta}_t, \sigma^2 \boldsymbol{I}) = \nabla_{\boldsymbol{\theta}_t} \log \frac{1}{\sigma \sqrt{2\pi}} \exp \left( \frac{(\boldsymbol{x} - \boldsymbol{\theta}_t)^2}{2\sigma^2} \right) \tag{10}$$

and by the linearity of the gradient operator, we can further simplify to:

$$\nabla_{\boldsymbol{\theta}_t} \log \frac{1}{\sigma \sqrt{2\pi}} \exp \left( \frac{(\boldsymbol{x} - \boldsymbol{\theta}_t)^2}{2\sigma^2} \right) = \nabla_{\boldsymbol{\theta}_t} \frac{(\boldsymbol{x} - \boldsymbol{\theta}_t)^2}{2\sigma^2} = \frac{(\boldsymbol{x} - \boldsymbol{\theta}_t)}{\sigma^2} = \frac{\boldsymbol{\epsilon}}{\sigma} \tag{11}$$

where the last step holds because $\boldsymbol{x} = \boldsymbol{\theta}_t + \sigma \boldsymbol{\epsilon}$, with $\boldsymbol{\epsilon} \sim N(\mathbf{0}, \sigma^2 \boldsymbol{I})$. Finally, after plugging this result into Equation (9), ES approximates the expected value of Equation (9) as the sample average of a population of $\lambda$ individuals $\boldsymbol{x}_i \sim N(\boldsymbol{\theta}_t, \sigma^2 \boldsymbol{I})$ (line 4):

$$\mathbb{E}_{\boldsymbol{x} \sim N(\boldsymbol{\theta}_t, \sigma^2 \boldsymbol{I})} \left[ f(\boldsymbol{x}) \nabla_{\boldsymbol{\theta}_t} \log N(\boldsymbol{\theta}_t, \sigma^2 \boldsymbol{I})(\boldsymbol{x}) \right] = \frac{1}{\sigma} \mathbb{E}_{\boldsymbol{\epsilon} \sim N(\boldsymbol{\theta}_t, \sigma^2 \boldsymbol{I})} [f(\boldsymbol{x}) \boldsymbol{\epsilon}] \approx \frac{1}{\sigma \lambda} \sum_{i=1}^{\lambda} f(\boldsymbol{x}_i) \boldsymbol{\epsilon}_i \tag{12}$$

## A.2  IMPLEMENTATION

We found some additions to the basic ThEBES algorithm to be useful in practice, though not algorithmically necessary.

First, we apply fitness shaping (introduced by Wierstra et al. (2014), but known since the 1970s) to the fitness values $s_i$ before computing the gradient at line 9 of Algorithm 2. Considering that we do not make any assumption about the fitness $f$ and the problem to be solved, the values $s_i$ could potentially lie in any range. As happens with other ESs (Wierstra et al., 2014; Hansen, 2016; Salimans et al., 2017; Chrabaszcz et al., 2018), fitness shaping makes the update invariant to the scale of the fitness, decreasing the probability of falling into local optima early and lowering the influence of outliers. Moreover, in our specific case, the stochastic drift $\boldsymbol{\eta}_t$ could "vanish" in the presence of unbounded $s_i$ values. To prevent this from happening, we apply the fitness shaping procedure of EvoJAX: it transforms the $s_i$ values into their ranks, which it then normalizes into $[-0.5, 0.5]$ (Salimans et al., 2017).

Second, we adopt symmetric sampling, similar to finite difference methods (Spall, 1998). Symmetric sampling is a common feature of contemporary ESs (Sehnke et al., 2010; Salimans et al., 2017) since it provides a more robust gradient approximation and we found it to improve convergence quality and time. Every time we pick a perturbation $\boldsymbol{\epsilon}_i$ from $N(\mathbf{0}, \boldsymbol{I})$, we add two symmetric samples $\boldsymbol{x}^+ = \boldsymbol{\theta}_t + \sigma \boldsymbol{\epsilon}_i$ and $\boldsymbol{x}^- = \boldsymbol{\theta}_t - \sigma \boldsymbol{\epsilon}_i$ to the population to be evaluated. Then, to have a population of $\lambda$ individuals at every iteration, we just need to sample $\frac{\lambda}{2}$ perturbations.

Third, we perform the gradient update using the state-of-the-art Adam scheme (Kingma & Ba, 2015), as it is appropriate for gradients that are very noisy or sparse. After preliminary experiments, we found it to improve convergence quality and time, in line with the findings of (Pagliuca et al., 2020) for policy search with ESs.

Finally, we decay both the learning rate $\alpha$ and the step-size $\sigma$ as evolution progresses; indeed, after preliminary experiments, we found that decaying both of them improved convergence quality and time over no decay or decaying only one of the two. At iteration $t$, the effective $\alpha_t$ and $\sigma_t$ are:

$$\alpha_t = \max(\tau^t \alpha_0, \alpha_{\text{limit}}) \tag{13}$$

$$\sigma_t = \max(\tau^t \sigma_0, \sigma_{\text{limit}}) \tag{14}$$

where $\tau \in [0, 1]$ is a decay rate, $\alpha_0$ is the initial learning rate, $\sigma_0$ is the initial step-size, and $\alpha_{\text{limit}}$ and $\sigma_{\text{limit}}$ are two minimum values.

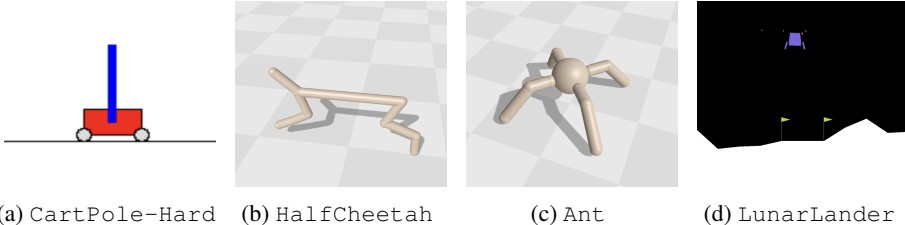

(a) `CartPole-Hard`   (b) `HalfCheetah`   (c) `Ant`   (d) `LunarLander`

Figure 6: The four policy search tasks we considered in our experiments.

### A.3 TASKS

We evaluate ThEBES on three policy search tasks for continuous control. We present a snapshot for each task in Figure 6.

`CartPole-Hard` consists of balancing a pole on a gliding cart (Barto et al., 1983), but it is a harder version of the classic benchmark: at the beginning of each simulation, it samples the observation from a uniform distribution in $[-1, 1]$ (the domain of the observations), thus injecting stochasticity in the environment. $r_k$ is the absolute difference between the actual pole angle and the $\frac{\pi}{2}$ angle, thus rewarding the agent for keeping the pole as upright as possible. The observation consists of the $x$-position of the cart, the $x$-velocity of the cart, the angle of the pole, and the angular velocity of the pole with respect to its upright standing, so $n_{\text{in}} = 4$. The action is the force to apply to the pole, so $n_{\text{out}} = 1$. A simulation terminates once $k_{\text{final}} = 1\,000$ time steps have elapsed or the cart drifts out of the screen, while a run terminates after $n_{\text{evals}} = 30\,000$ fitness evaluations. The size of the search space is $p = 4\,609$.

`HalfCheetah` and `Ant` are the Brax engine (Freeman et al., 2021) versions of the standard PyBullet (Coumans & Bai, 2016–2021) locomotion tasks. In both of them, the reward is the traveled distance over the $x$-direction. The observation consists of the $y$-position of the center of mass, the $x$-, $y$- and $z$-velocity, and the roll, pitch, yaw angles, and $(x, y, z)$ position (relative to the center of mass) of the robot's joints, so $n_{\text{in}} = 26$ and 28, respectively. The action consists of the torques to apply to the robot's joints, so $n_{\text{out}} = 6$ and 8, respectively. For both tasks, we set $k_{\text{final}} = 1\,000$ and $n_{\text{evals}} = 120\,000$. The sizes of the search spaces are $p = 5\,706$ and $p = 10\,312$, respectively.

`LunarLander` (Brockman et al., 2016) is, finally, a more challenging control task. The eponymous lander must adjust the throttle of the two engines to mantain a stable trajectory and safely land on the pad. The reward is inversely related to the distance from the landing pad, with bonus points for landing safely and a penalty for crashing. The observation consists of the $x$ and $y$ coordinates of the lander, its linear velocities, its angle, its angular velocity, and two booleans representing whether each leg is in contact with the ground or not, so $n_{\text{in}} = 8$. The action consists of the throttle for the two engines, so $n_{\text{out}} = 2$. We set $k_{\text{final}} = 1\,000$ and $n_{\text{evals}} = 120\,000$. The size of the search spaces is $p = 4\,866$.

For `CartPole-Hard`, `HalfCheetah`, and `Ant`, we relied on their EvoJAX (Tang et al., 2022) implementation, a recent GPU-accelerated evolutionary framework. For `LunarLander`, that is not available in EvoJAX, we relied on its gym (Brockman et al., 2016) implementation.

### A.4 HYPERPARAMETERS

For all of the baselines, we adopt the hyperparameter values suggested by Tang et al. (2022)—who carried out extensive grid search experiments for most of the algorithms and tasks considered—when available, otherwise on the values suggested by the respective authors; for `LunarLander`, we found the hyperparameter values for the Brax tasks to work well.

We summarize the hyperparameters set for ThEBES in Table 1. Unless otherwise specified, we adopt for ThEBES the same hyperparameter values suggested for OpenES, due to the similarity between the two algorithms.

| Name | Description | CartPole-Hard | HalfCheetah and Ant | LunarLander |
|------|-------------|---------------|---------------------|-------------|
| $\lambda$ | population size | 100 | 256 | 256 |
| $n_{\text{evals}}$ | number of fitness evaluations (as stopping criterion) | 30 000 | 120 000 | 120 000 |
| $\alpha_0$ | initial learning rate | 0.02 | 0.01 | 0.01 |
| $\sigma_0$ | initial step-size | 0.03 | 0.04 | 0.2 |
| $\tau$ | learning rate and step-size decay rate | 0.999 | 0.999 | 0.999 |
| $\alpha_{\text{limit}}$ | minimum learning rate | 0.001 | 0.005 | 0.005 |
| $\sigma_{\text{limit}}$ | minimum step-size | 0.01 | 0.001 | 0.001 |
| $\boldsymbol{\theta}_0$ | initial search distribution mean | **0** | **0** | **0** |

Table 1: Hyperparameter values for ThEBES.

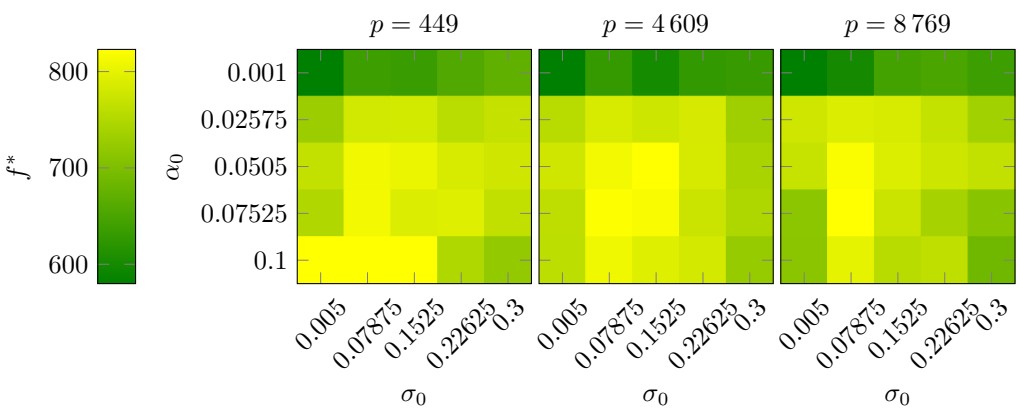

Figure 7: Heatmap for the median best fitness $f^*$ out of 5 independent runs on `CartPole-Hard`, for 3 numbers of hidden layers, 5 values for the initial learning rate $\alpha_0$, and 5 values for the initial step-size $\sigma_0$. Low values of $\alpha_0$ generally lead to lower fitness.

## A.5 IMPACT OF HYPERPARAMETERS ON THEBES

We study the impact of selected hyperparameters, in order to grasp how robust ThEBES is to their choice and have a picture of its overall behavior. We focus on population size $\lambda$, initial learning rate $\alpha_0$, and initial step-size $\sigma_0$.

### A.5.1 INITIAL LEARNING RATE $\alpha_0$ AND INITIAL STEP-SIZE $\sigma_0$

Since $\alpha_0$ and $\sigma_0$ jointly affect the smearing process, we jointly study their impact for different combinations. After preliminary experiments, we evaluate 5 equidistant values of $\alpha_0$ in $[0.001, 0.1]$ and 5 equidistant values of $\sigma_0$ in $[0.005, 0.3]$, while fixing $\lambda = 100$. We experiment on the `CartPole-Hard` task and, to understand how hyperparameter choices behave across different problem dimensionalities, we report the results for $p = 449$, $p = 4\,609$, and $p = 8\,709$ (corresponding to one, two, and three hidden layers, respectively). As a performance index, we consider the fitness of the best policy $f^*$ and report its median over 5 independent runs in Figure 7.

From the figure, we see that ThEBES is generally robust to the hyperparameters considered: performance never drops below $f^* = 600$ that, according to Tang et al. (2022), is the threshold for solving the `CartPole-Hard` task. At the same time, we spot some trends: very low values of $\alpha_0$ correspond to the worst performance (the dark green top rows), while higher values of $\alpha_0$ combined with lower values of $\sigma_0$ perform the best (the brighter colors of the lower-left quadrants). Higher values of $\alpha_0$ and lower values of $\sigma_0$ increase the relative impact of the gradient versus the smearing component, making evolution less "stochastic". This result suggests that low quantities of stochastic drift are beneficial when compared to none, while higher quantities are detrimental. Finally, there does not seem to be a significant impact of problem dimensionality on the performance of ThEBES. Considering that current evolutionary algorithms can evolve networks with millions of parameters

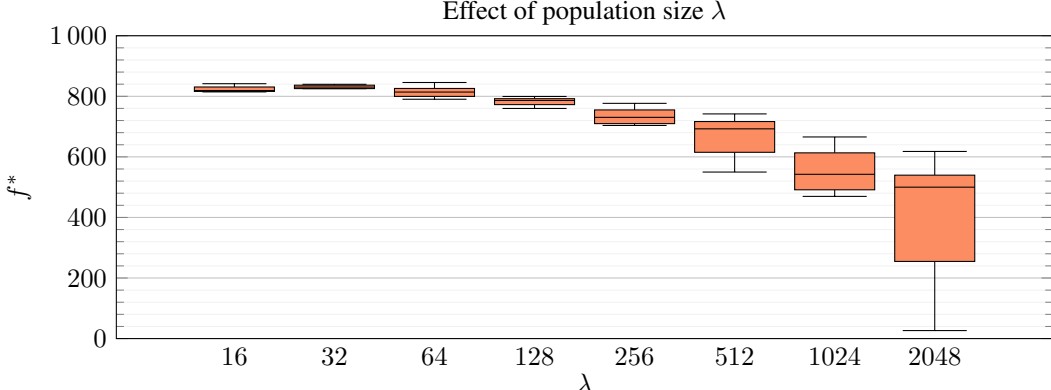

Figure 8: Distribution of the best fitness $f^*$ for 5 independent runs on `CartPole-Hard`, for 8 values of population size $\lambda$. Higher values of $\lambda$ lead to lower effectiveness.

(Stanley, 2007), we may need larger-scale experiments for this claim to be stronger; still, the network sizes considered insofar are reasonable for the policy search tasks that are common in the literature.

### A.5.2 POPULATION SIZE $\lambda$

To understand how population size impacts ThEBES, we evaluate $\lambda$ on an exponential scale, namely $\lambda = 2^k$ for $k \in \{4, 5, \dots, 11\}$. After preliminary experiments, we set $\alpha_0 = 0.02$ and $\sigma_0 = 0.03$ (their default values) to single out the impact of $\lambda$. We considered only the case of $p = 4\,609$. As a performance index, we consider the fitness of the best policy $f^*$ and report its distribution over 5 independent runs in Figure 8. Considering that all runs had the same computational budget of $n_{\text{evals}} = 30\,000$, higher values of $\lambda$ imply lower numbers of iterations.

From the figure, we see that higher values of $\lambda$ lead to lower performance: the higher $\lambda$, the lower $f^*$ on average. A lower number of iterations is probably a hindrance when following a stochastic path on the search space, as happens with ThEBES.

