# OpenReview forum: "ThEBES: Thorough Energy-Based Evolution Strategy"
_ICLR.cc/2024/Conference — ICLR 2024 Conference Withdrawn Submission_

### Official Review · Reviewer_NT4U · 2023-10-16

**Soundness:** 1 poor
**Presentation:** 2 fair
**Contribution:** 1 poor
**Rating:** 1
**Confidence:** 5

**Summary:**

The paper proposes a slight modification of an evolution strategy by adding noise to the distribution mean. The goal of this measure is to enhance robustness of the solution.

**Strengths:**

In machine learning, many newcomers seem to believe that neuroevolution started with the Salimans paper. I am glad to see citations of older work.

**Weaknesses:**

This work is based on a whole series of misconceptions. In the following, I restrict myself to a few central points. Actually, my list is still quite long.

The argumentation of this paper is based on the presumption that ESs promise robust solutions. I have been active in ES research for more than 10 years, and I don't remember anyone making such a promise.

Adding robustness to an optimization method is a very basic research topic in optimization. At its core, it is unrelated to machine learning. Therefore, the method should be evaluated on benchmark problems suited for assessing robustness of optimization. Evaluating on complex RL tasks does not allow me to judge whether the method achieves its goals, since the challenges of these problems are largely unknown. I would first like to see a clear-cut benchmark problem with robust optimization as the main challenge. Sphere and Rastrigin don't qualify for that goal. I really wonder why they were included.

Citing Wierstra et al. (2014) for the introduction of a population of fixed size and a Gaussian search distribution reveals that the authors' knowledge of evolution strategies (and their history) is shallow. That's a standard model since at least 40 years!

The subsequent formulas make things much worse. What is $\theta$? First, it refers to the vector of all distribution parameters, and that's also the meaning in the cited paper. From the next sentence onwards, $\theta$ is only the distribution mean, while the standard deviation (step size) \sigma is listed separately. The full covariance matrix considered by Wierstra is not even mentioned. The whole point of the NES method presented by Wierstra is to unify these parameters. What is presented here is the trimmed version sold by Salimans as an ES, although it does not even include the most basic of its mechanisms, namely step size adaptation. The manuscript does not even cite the OpenAI paper for this decisive difference. What happens here is that the term "ES" is misused to represent a stochastic gradient estimator from Gaussian samples which is then fed into ADAM. The evolutionary optimization community has a well-established (50 years old) definition of an ES. This method does not qualify.

The present paper takes a detour through the world of energy-based models to justify something trivial, namely adding noise to the distribution mean. I see no point in doing so, since the effect is very similar to using a larger $\sigma$. A so-called "ES" with fixed step size does not converge to a Dirac peak. Actually, it does not converge at all in the usual sense, but it approaches the limit distribution of a Markov chain that usually (and hopefully) covers high quality solutions without actually identifying them more precisely. Then why should one need to add noise to the mean in order to achieve the stated goal of the paper? One could simply increase $\sigma$ to get essentially the same effect.

The fitness plots for Sphere and Rastrigin do not use a log scale. This way, they are completely meaningless. That's entirely unacceptable and ignorant of the standards of the field. Also, the curves for CMA-ES clearly reveal a problem. Here, CMA-ES is slower than random search! In reality, CMA-ES is extremely efficient on Sphere. The only explanation might be that it is initialized with an unsuitable step size in combination with a high problem dimension. It is also compelling that the curves for Sphere and Rastrigin look identical. That should really not be that case. I wonder how one can publish such results without being a bit more skeptical. The authors neither state the problem dimension, nor the initialization, and not even the computational budget. However, one thing is crystal clear: no algorithm in the world can outperform CMA-ES on sphere by a large margin, because it is actually quite efficient on that problem. Since the code is not included as a supplement, I am unable to give a more precise analysis. I can however conclude with certainty that the results cannot be trusted. They are as fishy as they can get.

From the appendix: "Finally, we decay both the learning rate \alpha and the step-size \sigma as evolution progresses; indeed, after preliminary experiments, we found that decaying both of them improved convergence quality and time over no decay or decaying only one of the two." I strongly recommend the authors to read (a translation of) the 1973 Rechenberg paper they cite. It is all about adapting \sigma properly, and when implemented (and using fitness shaping as discussed in the appendix), there will be no need for any adaptation schedule, which truly adds lots of hidden hyperparameters.

Further problems:

The "Separable Natural Evolution Strategy (sNES)" was not published by Wierstra (as claimed), but by Cuccu and Gomez:
Cuccu, Giuseppe, and Faustino Gomez. "Block diagonal natural evolution strategies." International Conference on Parallel Problem Solving from Nature. Berlin, Heidelberg: Springer Berlin Heidelberg, 2012.

In the experiments, CMA-ES is applied to networks with more than 4000 weights. With that problem dimension, it takes extremely long to adapt the covariance matrix. In essence, it will not happen within the given budget. It would be much more sensible to apply a low-rank approximation method like here:
Loshchilov, Ilya. "A computationally efficient limited memory CMA-ES for large scale optimization." Proceedings of the 2014 Annual Conference on Genetic and Evolutionary Computation. 2014.

Section numbering: 2 -> 2.0.1 -> 2.1 ... something is wrong here.

The paper contains unresolved LaTeX references ("??").

**Questions:**

I would be very much interested in the experimental settings that make CMA-ES perform worse than random search on the Sphere problem.
Also, which implementation of CMA-ES was used?

---

### Official Review · Reviewer_Ew2x · 2023-10-23

**Soundness:** 2 fair
**Presentation:** 3 good
**Contribution:** 2 fair
**Rating:** 3
**Confidence:** 4

**Summary:**

Evolution Strategies (ESs) have recently shown significant advancements in evolutionary algorithms, aiming to generate robust solutions. However, the existing methods fall short in delivering this promise. The proposed Thorough Energy-Based Evolution Strategy (ThEBES) introduces noise to the search distribution mean, employing Langevin dynamics grounded on Energy-Based Models (EBMs) to converge to a probability distribution. ThEBES demonstrates competitive performance and increased robustness to observational noise compared to other ESs, highlighting its potential for future research and theoretical advancements in the field.

**Strengths:**

This paper proposes an energy-based evolutionary strategy (ThEBES) that increases the robustness and interpretability of ES.　No hyperparameters need to be added to ThEBES.

**Weaknesses:**

1. This paper adds random drift to ES through Langevin dynamics, which improves the robustness of ES. However, this paper is not innovative enough, similar ideas (increasing the diversity and robustness of algorithm) are present in many studies.
2.This paper chooses five algorithms as its baseline, but lacks the latest algorithms, including, e.g.:
1)BCMA-ES: A Bayesian approach to CMA-ES.
2)Maxmin Q-learning: Controlling the Estimation Bias of Q-learning.
3)Sample and time efficient policy learning with CMA-ES and Bayesian Optimisation.
4)Toward a Matrix-Free Covariance Matrix Adaptation Evolution Strategy.
3. There are some errors or unclear presentation in the paper, such as (see ??) in the page 6.
4. In the last few years, there have been a lot of papers related to the EA theory, such as:
1)Theory of (1+ 1) ES on SPHERE revisited.
2)Probabilistic Analysis of the (1+1)-Evolutionary Algorithm.
The authors' literature review lacks comprehensiveness.

**Questions:**

1.Some similar ideas (increasing the diversity and robustness of the algorithm) are present in many studies. Mathematically interpretable ES have also been studied a lot, such as some improved CMAES. The innovation of THEBES is not outstanding enough.
2.This paper chooses five algorithms as its baseline, but lacks the latest algorithms, for example:
1)BCMA-ES: A Bayesian approach to CMA-ES.
2)Maxmin Q-learning: Controlling the Estimation Bias of Q-learning.
3)Sample and time efficient policy learning with CMA-ES and Bayesian Optimisation.
4)Toward a Matrix-Free Covariance Matrix Adaptation Evolution Strategy.
The authors should add some comparative algorithms.

---

### Official Review · Reviewer_HLqH · 2023-11-01

**Soundness:** 1 poor
**Presentation:** 1 poor
**Contribution:** 1 poor
**Rating:** 3
**Confidence:** 5

**Summary:**

This paper proposes a variant of evolution strategy based on the one often referred to as OpenAI-ES. The modification is very simple. It simply added a noise term to the parameter update. The proposed approach, ThEBES has been evaluated on 6 test problems including 2 typical benchmark problems and 4 control tasks.

**Strengths:**

A novel and simple modification of ES has been proposed.

**Weaknesses:**

Very limited novelty. The algorithm only add the noise term to the mean vector update. However, such an effect is naturally included in the original update of the mean vector in ES as as it takes the weighted average of the candidate solutions, which results in the expected gradient with an additional gaussian noise scaled by the step-size. Therefore, I do not see any technical advantage of the proposed algorithm.

The experimental results are rather suspicious. On Sphere and Rastrigin, the proposed approaches and some other approaches reached better solutions already at the beginning than the solutions that CMA-ES reached at the end of the search. It is definitely due to unfair and invalid initialization and hyper-parameter settings.

The proposed approach is proposed as a general purpose black-box optimization algorithm. For such a purpose, the numerical evaluation is definitely not sufficient. Consider to use COCO (BBOB) framework to evaluate the algorithm and to compare with other baselines.

The robustness is evaluated in the paper, but if the robustness is an important point, there are many approaches to obtain robust policies. The authors should compare the proposed approach to these approaches.

The explanations of ES are wrong. In Wierstra 2014, they parameterized both the mean and the covariance of the gaussian distribution, not just the mean. Algorithm 1 is wrong. The epsilon should follow N(0, 1), not N(0, sigma^2).

**Questions:**

Please answer to the points mentioned above.

---

### Official Review · Reviewer_Qw4z · 2023-11-08

**Soundness:** 2 fair
**Presentation:** 1 poor
**Contribution:** 2 fair
**Rating:** 3
**Confidence:** 3

**Summary:**

This paper has introduced the energy-based model in Evolution Strategies by adding noise in the search distribution for robustness and a stochastic drift to the ES search trajectory.

**Strengths:**

This paper has proposed an evolutionary algorithm with the interpretation from energy-based models. The proposed method achieves robust performance with the search method based on Langevin dynamics in the policy space.

**Weaknesses:**

This paper has introduced a noise term for parameter estimation within Evolution Strategies (ESs). However, the underlying issue appears more akin to an optimization challenge where a broader exploration of stochastic gradient descent (SGD) techniques should be included and compared in both the discussion and experiments. Notably, the novelty seems somewhat limited, given the extensive prior research on the application and enhancement of SGD in policy optimization, as demonstrated in works such as [1][2][3]. Aside from these points, there is a need for a thorough reorganization of the content, as I am finding it difficult to discern the specific problem that ES aims to address. Please refer to my detailed comments below for further clarification.

1. I believe the author's intent is to introduce a policy search algorithm within the context of reinforcement learning tasks. Nevertheless, for individuals who lack familiarity with this subject matter, grasping the connection between ES (Evolution Strategies) and the policy search algorithm can be challenging when perusing the introduction.
2. The paper overlooks citing the pioneering work on Energy-Based Models (EBMs) utilizing a neural network as the energy function in the generative model, as presented in [4].
3. The section number 2.0.1 should be 2.1.
4. The author claims that "The energy is a smoothing of the fitness function that makes the fitness landscape smoother and hence better behaved". Is there any evidence or experiments can demonstrate this?
5. In section 4, there is a "??" in reference.

[1] Lu, Songtao, et al. "Decentralized policy gradient descent ascent for safe multi-agent reinforcement learning." Proceedings of the AAAI Conference on Artificial Intelligence. Vol. 35. No. 10. 2021.

[2] Massaroli, Stefano, et al. "Learning stochastic optimal policies via gradient descent." IEEE Control Systems Letters 6 (2021): 1094-1099.

[3] Pham, Nhan, et al. "A hybrid stochastic policy gradient algorithm for reinforcement learning." International Conference on Artificial Intelligence and Statistics. PMLR, 2020.

[4] Xie, Jianwen, et al. "A theory of generative convnet." International Conference on Machine Learning. PMLR, 2016.

**Questions:**

1. What is population refer to in the model? Is it a reward function in RL problems?
2. Could the author explain what problem the ES try to solve? Is there any realistic example?
3. In 2.1, what does it mean for "$\theta$ is $\theta$"?
4. Is $x$ also an variable in Eq(3) for the numerator?
5. Why the solution space follows Gaussian distribution?
6. What does the introduced drift term stands for in the proposed framework. Is it used only for a stochastic gradient descent method?
7. Could the proposed method be used in discrete space?